# Study on the Constitutive Equation and Mechanical Properties of Natural Snow under Step Loading

Hongwei Han [1,2], Meiying Yang [1], Xingchao Liu [1], Yu Li [1], Gongwen Gao [1] and Enliang Wang [1,2,*]

1   School of Water Conservancy and Civil Engineering, Northeast Agricultural University, Harbin 150030, China; hanhongwei@neau.edu.cn (H.H.); 18346271554@163.com (M.Y.); dnlxc@neau.edu.cn (X.L.); 18846771809@163.com (Y.L.); 18045129438@163.com (G.G.)
2   Heilongjiang Provincial Key Laboratory of Water Resources and Water Conservancy Engineering in Cold Region, Northeast Agricultural University, Harbin 150030, China
*   Correspondence: hljwel@126.com

**Abstract:** Snow, as an important component of the cryosphere, holds a crucial role in the construction of polar infrastructure. However, the current research on the mechanical properties of snow is not comprehensive. To contribute to our understanding of the mechanical behaviors of snow in cold regions, uniaxial compression tests under step loading were performed on the snow. With the Maxwell model as the basis, different temperatures, densities, and loading rates were set to establish constitutive equations of snow. The changes in the elastic modulus and viscosity coefficient of snow with respect to three variables were investigated. The results show that the loading rate has no obvious effect on the elastic modulus and viscosity coefficient of snow. Both the elastic modulus and viscosity coefficient of snow follow an exponential function with respect to density, with an increase in density, resulting in a higher value. As temperature decreases, the elastic modulus and viscosity coefficient initially decrease and then increase, whereas no specific functional relationship between them was observed. Additionally, a new constitutive equation considering snow density is derived based on the Maxwell model.

**Keywords:** snow; step loading; constitutive equation; Maxwell model

## 1. Introduction

Snow affects the replenishment of water resources, the occurrence of natural disasters, and the changes in air quality [1]. Approximately 98% of the global seasonal snow cover is found in the Northern Hemisphere [2], of which about 60–65% is in Europe and Asia [3]. Extreme snowfall often occurs in the French Alps due to geographical location [4]. In the western United States, snowfall serves as a major source of domestic and agricultural water supply [5]. Similarly, winter snowfall provides water recharge in arid regions of Iran [6]. Snowfall in Southern China increases in winter due to the presence of the Tibetan Plateau, which leads to high relative humidity and discomfort in the south [7]. In recent years, the polar route has attracted significant interest as increasingly more countries have set up scientific research stations in Antarctica for observation and research. Given the extremely cold weather and geographical location of the Antarctic region, the mode of delivering supplies and personnel to these research stations is limited to sea and air transportation. However, sea transportation is time-consuming and subject to seasonal restrictions, leading many countries to initiate airport construction in Antarctica to facilitate research. According to the construction location and runway type, they can be categorized as sea ice runways, blue ice runways, sled runways, and compacted snow runways [8,9]. Challenges faced by the first three types of airport runways include high construction costs, limited available areas, and insufficient snow-layer strength. In contrast, compacted snow runways make full use of Antarctic snow resources and offer low maintenance costs, becoming the preferred choice for airport construction in many countries [10]. Nevertheless,

existing research on snow mechanics remains incomplete due to the spatial anisotropy of snow and its continuous metamorphism over time [11,12]. Consequently, analyzing the deformation behavior and mechanical properties of snow is of great interest when it comes to engineering construction.

Snow is composed of three phases of water, air, and ice connected together in the form of particles, and has a certain strength [13]. Subsequently, many experimental studies on snow have been conducted and various constitutive equations have been developed to understand snow deformation under loading conditions [14,15]. At present, the constitutive equations of snow can be divided into two kinds. One is to describe the deformation characteristics of snow from a macroscopic perspective, which is also called the phenomenological method. The other takes a microscopic viewpoint by observing the changes in ice particles and bonds within the snow under the external loads and then deducing the overall deformation.

Mishra and Mahajan [14] considered that snow deformation consists of elastic deformation and creep deformation, ignoring time-hardening and microstructural changes. They described the constitutive relationship of snow based on a complementary power potential, which also predicted volume changes in snow samples. Snow can be regarded as a geotechnical material whose internal microstructure determines the overall stress–strain relationship. Birkeland et al. [16] and Bobillier et al. [17] developed discrete element models to simulate the propagation saw test, aiming to understand the microstructural changes during snow failure. Nicot [18] assumed that the mechanical behavior of snow primarily depends on the mechanical properties of internal bonds. They constructed a probability density function to describe the microstructure of snow. Using a nonlinear Kelvin model and a fabric description as constitutive equations, the overall mechanical behavior is deduced from the local properties of snow. Mahajan and Brown [19] constructed a multi-axial constitutive equation for snow, which pointed out that the deformation of snow was divided into different mechanisms. The deformation of the entire snow sample under external load was deduced based on this determined mechanism. Brown [20] constructed a volume constitutive equation of snow based on the changes in bond size and ice grain diameter. However, the accuracy of the equation is reduced at high snow densities because the interactions between adjacent bonds are not considered. Recent efforts have aimed to combine tomography technology with the discrete element method to build microscopic numerical models of snow and simulate its deformation behavior under load [21–24]. Singh et al. [25] assumed snow to be an orthotropic elastoplastic material and constructed an equation to predict the constitutive relationship of snow with different densities and types. X-ray microtomography combined with the finite element method was used to determine the parameters in the constitutive equation and verify the results. Chandel et al. [26] determined the deformation of ice particles under load based on the damage elastic–plastic constitutive equation of ice. They used X-ray tomography technology and finite element software to simulate the stress–strain relationship of RVE and derived the overall macroscopic constitutive behavior of snow. From the perspective of the relationship between airflow and compaction, a compacted snow constitutive model was proposed under confined compression tests. This model was consistent with laboratory measurements, indicating that air inside the snow sample would be discharged from the pores during testing [27]. During the cold winter months, rivers and lakes begin to freeze and snowfall occurs. The abundant snow and ice resources are often used as building materials. Such as the utilization of ice and snow for constructing airstrips in the Antarctic region [28], creating snow and ice sculptures, constructing distinctive buildings, and the construction of snow and ice roads to facilitate transportation on ice [29]. In construction, snow needs to be crushed and other operations need to be carried out to improve its strength to ensure the safety of buildings and roads. The relevant features of the mechanical behavior of snow need to be fully understood, and constitutive equations should be used to describe the stress–strain relationship of snow.

In previous studies, the constitutive model of snow is frequently investigated using relative theoretical derivations or numerical simulation methods [14,26], and then verified by experimental data. However, the numerous parameters and complex formulas involved in these methods make them difficult to compute. Questions still remain regarding the accurate interpretation of the deformation behavior of snow. In this paper, the constitutive equations of snow obtained by combining laboratory tests and theoretical derivations can more truly reflect the mechanical behavior of snow, which is an extension and supplement to the previous research. In general, the main objective of this paper is to provide a novel constitutive model aimed at representing the most relevant aspects of the overall macro-behavior of snow while retaining adequate simplicity so that a feasible application is achieved in practical (large-scale, long-duration) engineering cases. The derivation method of the constitutive law for solid materials is used to obtain the constitutive law of snow. This study focused on investigating the constitutive equation and mechanical behavior of snow by conducting step-loading uniaxial compression tests on cylindrical snow specimens. Sections 2 and 3 detail the preparation of the snow samples and the uniaxial compression test procedure. After some preliminary notes on the Maxwell model, the derivation process of the elastic modulus and viscosity coefficient through it is introduced. The changes in the elastic modulus and viscosity coefficient of snow in response to changes in snow density, loading rate, and temperatures are studied and discussed in Section 4. A new constitutive equation is proposed to link the elastic modulus and viscosity coefficient of snow to its density, temperature, and loading rate. Section 5 discusses the trends of elastic modulus and viscosity coefficient obtained in this paper with respect to different variables in comparison with the results of other studies in the literature. Finally, the major results and implications of this paper are emphasized in the conclusion.

## 2. Sample Preparation and Experimental Procedure

### 2.1. Sample Preparation

The snow samples used in this study are fresh natural snowfall from the top layer of the ground at the Water Conservancy Comprehensive Test Site of Northeast Agricultural University in Harbin, Heilongjiang Province, China. As one of the three main snow areas in China, low temperatures and abundant snow are the primary characteristics of the northeastern region during winter due to the temperate monsoon climate and high latitude. In general, the main snowfall period is from October to March of the following year, and the stable snowfall period is from November to February of the following year. Commonly, fresh snowfall grains are loose and uniform in size, without any bonding for easy compaction. Referring to the snow sample preparation method of [30], from 26 November to 1 December 2022, fresh snow was collected and poured into the compaction equipment by manual layering to achieve uniform density and to minimize errors. The compaction instrument consists of a guard cylinder, a compaction hammer, and a compaction cylinder to compact the snow into a 100 mm diameter, 200 mm high cylinder (height-to-diameter ratio of 2:1), as depicted in Figure 1. It is provided by Suzhou Tuo Testing Instrument Equipment Co., Ltd. located in Suzhou, Jiangsu Province, China. Snow samples with densities of $350 \, \text{kg/m}^3$, $400 \, \text{kg/m}^3$, $450 \, \text{kg/m}^3$, $500 \, \text{kg/m}^3$, and $550 \, \text{kg/m}^3$ were created using different masses of snow. To prevent the snow samples from sublimating during storage and to maintain integrity, they were wrapped tightly in cling film and stored in a refrigerator at a constant temperature of $-25 \, ^\circ\text{C}$. As the tests were carried out at different temperatures, the snow samples were stored in a refrigerator at the required temperature for 24 h prior to each test to ensure uniform temperature throughout the entire sample. Before the test began, the mass and height of the snow samples were measured using an electronic scale and tape measure to calculate the actual density, ensuring that the error between it and the experimental density was less than $\pm 20 \, \text{kg/m}^3$. Through measuring the mass and volume of snow, the natural density of fresh snow was derived to be $128 \, \text{kg/m}^3$. The above operations can reduce the error to the tolerable range.

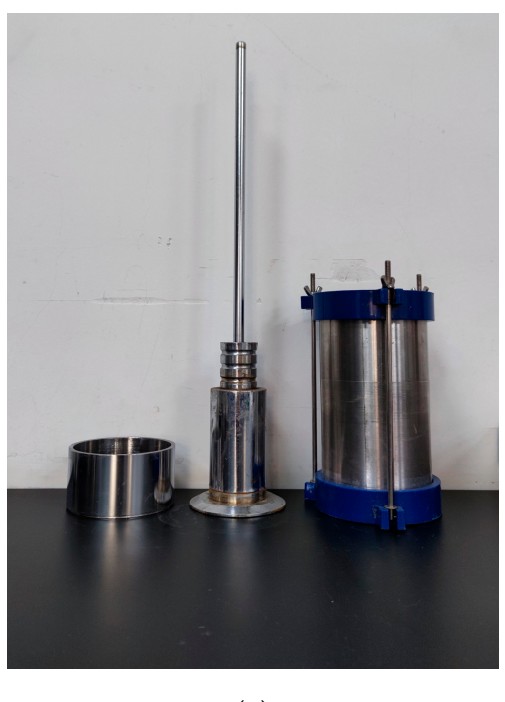

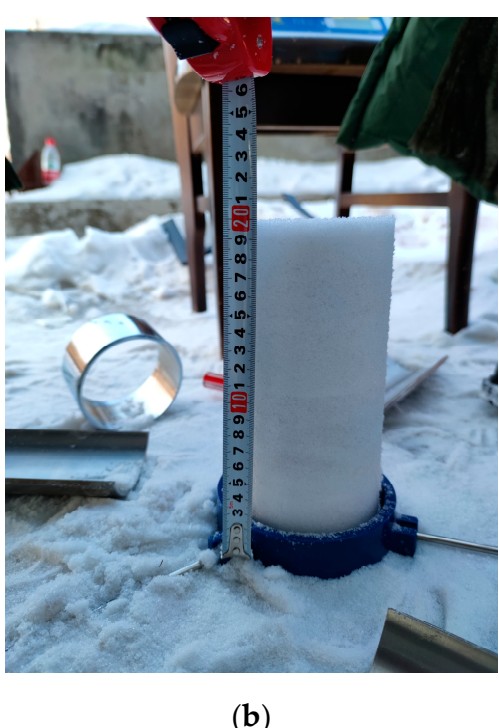

(**a**)          (**b**)

**Figure 1.** Compacting tools and snow sample: (**a**) Compaction equipment, from left to right, the protective cylinder, the compactor hammer, and the compaction cylinder are followed; (**b**) sample.

### 2.2. Test Procedure

The uniaxial compression tests were carried out on snow samples using a load-controlled step loading method at temperatures of $-5$ °C, $-10$ °C, $-15$ °C, and $-20$ °C, with loading rates of 5 N/s, 10 N/s, 20 N/s, 30 N/s, 40 N/s, and 50 N/s, respectively. The force difference between the two adjacent steps was set to 300 N, and each step lasted for more than 200 s. As shown in Figure 2, the experimental equipment in this study includes a WDW–100 electronic universal testing machine and two infrared sensors. The WDW–100 machine is made up of a compression device, a control system, and a low-temperature test chamber. It comes from Changchun Kexin Testing Instrument Co., Ltd. located in Changchun city, China. The low-temperature test chamber is used to control the temperature inside the testing machine, so that the snow sample can be tested at a preset temperature, with a minimum temperature of $-30$ °C. Two infrared sensors were placed inside the testing machine to record the diameter expansion of the snow sample in real-time during compression. One sensor was located near the middle of the snow sample, and the other near the end of the snow sample. Before the test, the upper and lower pressure plates of the testing machine were wrapped with cling film to avoid the end effect impacting the test results during compression. Furthermore, upon setting the internal temperature of the testing machine to the desired test temperature, the sample was put into it when the internal temperature was uniform. During the test, the central axis of the specimen was aligned in a straight line with the center of the upper and lower pressure plates to ensure uniform force distribution. To minimize errors in the recorded diameter growth value, the infrared light emitted by the sensor was directed as closely as possible to the center of the snow sample.

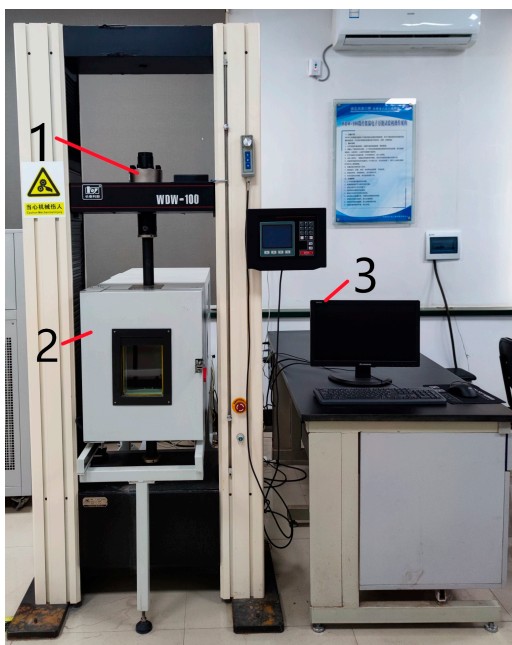

**Figure 2.** WDW–100 universal testing machine: 1. compression device; 2. cryogenic test chamber; 3. control system.

## 3. Constitutive Model

### 3.1. Maxwell Model

The stress–strain curve is a complete macroscopic response to a series of change processes such as the deformation of snow under external force, the rearrangement of internal grains, and the breakage and peeling of the exterior. The constitutive equation is the mathematical expression describing this curve, which is the main basis for studying and analyzing snow's bearing capacity and deformation. There are two main methods to establish the constitutive model of crystalline materials, at present. One of which is to describe the constitutive equation of materials by a series-parallel combination of elastic, viscous, and plastic elements. The other is to synthesize empirical formula to describe the constitutive equation of materials according to the stress–strain change law obtained from experiments. This study adopts the method of component combination to construct the constitutive equation of snow. The widely used classic constitutive models currently include the Maxwell model, Burgers model, and Kelvin model. This study assumes that snow is a kind of viscoelastic material and chooses the Maxwell model as the basis. Three constitutive equations are derived to describe the mechanical behavior of snow to show the effects of temperature, density, and loading rate on the viscoelasticity of snow.

The Maxwell model comprises a Hooke body and a Newtonian body connected in series, as illustrated in Figure 3. When stress is applied to the snow, both elastic strain and plastic strain are generated. The Hooke body represents the instantaneous elastic deformation that occurs upon the application of stress, which can be recovered immediately after the stress is removed. In contrast, the Newtonian body represents the viscous behavior of the material, which produces irreversible deformation under stress. After the stress is unloaded, the deformation remains. A brief description of the Maxwell model is presented as follows.

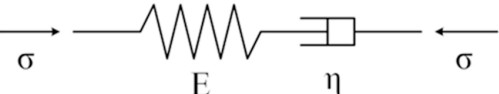

**Figure 3.** Maxwell model.

The total strain consists of elastic and viscous strains:

$$\varepsilon = \varepsilon_E + \varepsilon_\eta \tag{1}$$

Instantaneous elastic strain is inversely proportional to the elastic modulus, $E$:

$$\varepsilon_E = \frac{\sigma_E}{E} \tag{2}$$

Viscous strain rate is inversely related to the viscosity coefficient, $\eta$:

$$\dot{\varepsilon}_\eta = \frac{\sigma_\eta}{\eta} \tag{3}$$

When the elements are connected in series, they are subjected to the same stresses:

$$\sigma = \sigma_E = \sigma_\eta \tag{4}$$

Substituting these relationships (Equations (2)–(4)) into Equation (1), the constitutive equation is derived:

$$\dot{\varepsilon} = \dot{\varepsilon}_E + \dot{\varepsilon}_\eta = \frac{\dot{\sigma}}{E} + \frac{\sigma}{\eta} \tag{5}$$

Integrating Equation (5) over time, the deformed coordination equation is obtained as:

$$\varepsilon = \varepsilon_E + \varepsilon_\eta = \frac{\sigma_E}{E} + t\frac{\sigma_\eta}{\eta} = \sigma\left(\frac{1}{E} + \frac{t}{\eta}\right) \tag{6}$$

*3.2. Parameter Calculation*

The stress of the snow sample under the uniaxial compression test with step loading remains constant in the horizontal step when the stress rate is 0. Substituting this condition into Equation (5), the expression for the viscosity coefficient of snow can be obtained as:

$$\eta = \frac{\sigma}{\dot{\varepsilon}} \tag{7}$$

The elastic modulus can be obtained using Equation (6):

$$E = \frac{1}{\left(\frac{\varepsilon}{\sigma} - \frac{t}{\eta}\right)} \tag{8}$$

where $\sigma$ is the constant stress at the step (MPa); $\varepsilon$ is the strain; $\dot{\varepsilon}$ is the strain rate (s$^{-1}$); $t$ is the time (s); $E$ is the elastic modulus of snow (MPa); and $\eta$ is the viscosity coefficient of snow (MPa·s).

## 4. Results

The stress–strain curves of the snow samples obtained by step loading are shown in Figure 4. The uniaxial compression test under step loading is completed alternately by the normal uniaxial compression test and creep test. Indeed, the data of the first and last steps are not involved in the calculation of parameters. Because there had been a pre-pressure adjustment in the first stage, the measured displacement value is greater than the actual deformation, resulting in a relatively small and inaccurately calculated viscosity coefficient. In the final step stage, as the density of the snow sample gradually increases, the bonds between the ice particles begin to deform, break, and rearrange, which cause an increase in deformation of the snow sample under loading. The calculated viscosity coefficient is also relatively small. Due to the low density and the larger internal pores of snow compared to brittle materials such as rocks and ice, the stress of the snow sample under load will rapidly decrease after reaching a set value at the beginning of the step. At this point, the

pores inside the snow collapse, and the ice crystals begin to rearrange and combine. When a relatively stable structure is reached, the stress will stop decreasing and enter a creep state as shown in Figure 4. The elastic modulus and viscosity coefficient of compacted snow under step loading are then calculated using Equations (7) and (8) and the variation of these mechanical properties are investigated in relation to temperature, density, and loading rate.

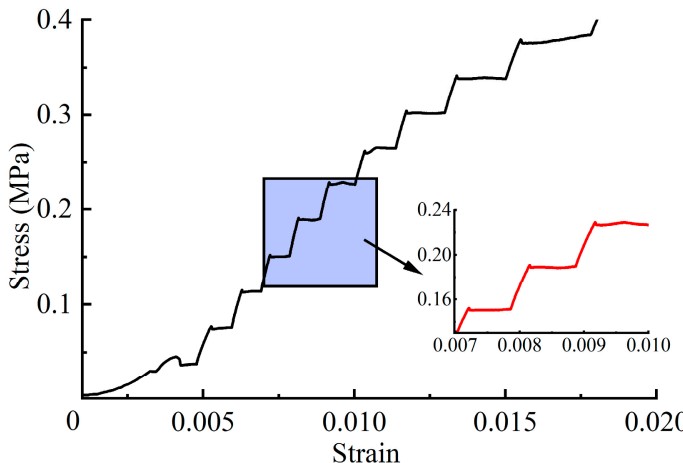

**Figure 4.** Stress–strain curves of 550 kg/m$^3$ snow samples at $-15$ °C and 10 N/s.

### 4.1. Effect of Loading Rate on Elastic Modulus and Viscosity Coefficient of Snow

Uniaxial compression tests were carried out on 400 kg/m$^3$ and 500 kg/m$^3$ snow samples at a constant temperature of $-15$ °C with step loading rates ranging from 5 to 50 N/s to observe the effect of loading rate on the elastic modulus and viscosity coefficient. As shown in Figure 5a, the elastic modulus is more scattered for the 500 kg/m$^3$ snow than for the 400 kg/m$^3$ snow. With increasing loading rate, the value in snow elastic modulus is significantly greater at high densities than at low densities, which indicates that the elastic modulus gradually increases with increasing density. At the same time, it is found that density is a factor that affects the degree of change in elastic modulus, with a greater impact on the elastic modulus with loading rate at higher densities. For example, the elastic modulus of the 400 kg/m$^3$ snow with a range of 5 to 50 N/s is 3.39 to 9.33 MPa, and the difference is 5.94 MPa. The elastic modulus of the 500 kg/m$^3$ snow with a range of 5 to 50 N/s is 7.18 MPa to 26.98 MPa, and the difference is 19.80 MPa. However, the results for the elastic modulus of snow show no clear tendency for dependence on the loading rate.

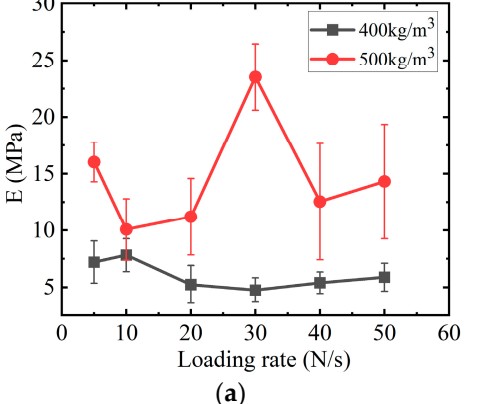 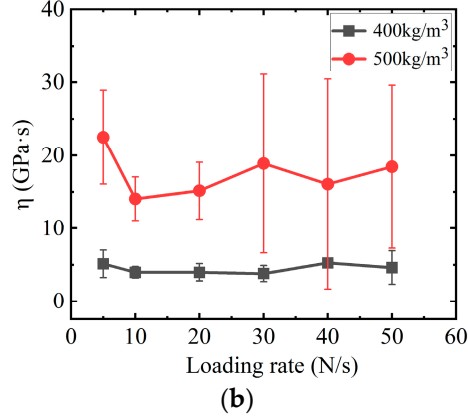

**Figure 5.** The relationship between the viscoelastic properties and loading rate: (**a**) the elastic modulus; (**b**) the viscosity coefficient.

Figure 5b compares the viscosity coefficient versus loading rate for two density snow samples at −15 °C temperature. The viscosity coefficient varies from 3.26 to 32.98 GPa·s for the 500 kg/m$^3$ snow sample and from 2.57 to 7.27 GPa·s for the 400 kg/m$^3$ snow sample, showing that the results are more scattered for the high-density snow. The average value of the viscosity coefficient for 500 kg/m$^3$ snow is larger than for 400 kg/m$^3$ snow, which suggests that the viscosity coefficient tends to increase gradually with an increase in density. Density also influences the change degree of the viscosity coefficient from Figure 5b. The difference in viscosity coefficients is 4.70 GPa·s for snow samples at 400 kg/m$^3$, and 29.72 GPa·s for snow samples at 500 kg/m$^3$. It is shown that the higher the density, the more intense the change in viscosity coefficient. When the loading rate is less than 10 mm/min, the mean values of the viscosity coefficients at both densities tend to decrease with an increase in loading rate. However, the mean values of viscosity coefficients at the two densities show different trends with increasing loading rate. The mean value of snow viscosity coefficient tends to be a constant at 400 kg/m$^3$, while the mean value of viscosity coefficient tends to increase, then decrease, and then increase again at 500 kg/m$^3$.

### 4.2. Effect of Temperature on Elastic Modulus and Viscosity Coefficient of Snow

As shown in Figure 6a, the change in elastic modulus of snow samples with different densities is shown for various temperatures (−5 °C, −10 °C, −15 °C, and −20 °C) under a loading rate of 10 N/s. A trend is observed at temperatures of −15 °C and −20 °C, where the elastic modulus increases as the density increases. However, at −5 °C and −10 °C, the average value of elastic modulus for the 550 kg/m$^3$ snow samples is lower than that of the 500 kg/m$^3$ snow samples, which reverses the previous results. It is found that the average value of the elastic modulus first decreases and then increases with decreasing temperature above 500 kg/m$^3$ and fluctuates in a very narrow band with a density at 400 kg/m$^3$. This concludes that the temperature has an impact on the change of the elastic modulus in the density range above 500 kg/m$^3$. However, more experiments need to be conducted at densities below 500 kg/m$^3$ to uncover the exact density range within which temperature has an impact.

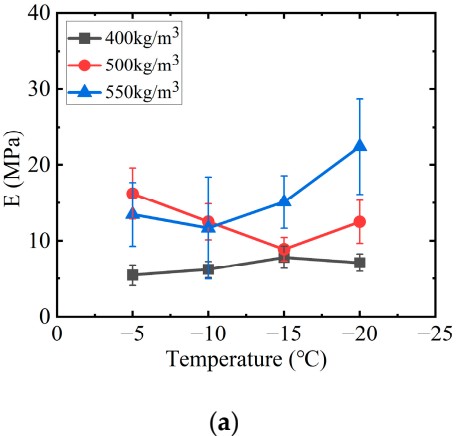
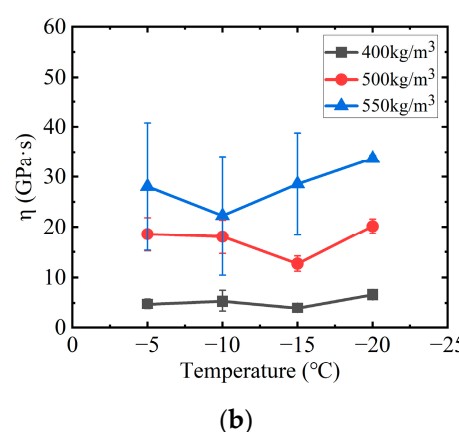

(**a**)　　　　　　　　　　　　　　　　　　　　　　　　(**b**)

**Figure 6.** The relationship between the viscoelastic properties and temperature: (**a**) the elastic modulus; (**b**) the viscosity coefficient.

As shown in Figure 6b, the variation of viscosity coefficient with temperature is shown for snow samples at each density. The dispersion degree of viscosity coefficients is lower for low-density snow samples and higher for high-density snow samples. At the density of 400 kg/m$^3$, the viscosity coefficient of snow varies from 3.36 to 7.77 GPa·s, and the difference is 4.41 GPa·s. At the density of 500 kg/m$^3$, the viscosity coefficient of snow varies from 11.04 to 21.78 GPa·s, and the difference is 10.74 GPa·s. The mean value of the viscosity coefficient generally decreases and then increases with decreasing temperature. Furthermore, it is noted that the density of the snow samples affects the degree of variation in the mean

viscosity coefficient with temperature. We also analyze the changes in the average value of viscosity coefficient under three densities. It is found that the minimum viscosity coefficient of snow at the density of 400 kg/m$^3$ is 3.98 GPa·s at −15 °C, and the maximum value is 6.64 GPa·s at −20 °C, with a difference of 2.66 GPa·s. The minimum viscosity coefficient of snow at the density of 500 kg/m$^3$ is 12.71 GPa·s at −15 °C, the maximum value is 20.13 GPa·s at −20 °C, and the difference is 7.42 GPa·s. Similarly, the minimum viscosity coefficient of snow at a density of 550 kg/m$^3$ is 22.24 GPa·s at −10 °C, and the maximum value is 33.68 GPa·s at −20 °C, with a difference of 11.44 GPa·s. It can be observed that from the above results, the viscosity coefficient of high-density snow samples fluctuates significantly more than that of low-density snow samples as the temperature decreases. Previous research results have shown that the viscosity coefficient can vary with temperature by four to five orders of magnitude, which was concluded via uniaxial compression tests [31].

### 4.3. Effect of Density on Elastic Modulus and Viscosity Coefficient of Snow

Figure 7 shows the effect of density on the elastic modulus and viscosity coefficient at constant temperature and loading rate. It should be noted that the elastic modulus of compacted snow increases as the density increases, which is consistent with the trend obtained by the authors of [32]. The elastic modulus of snow with the same density is relatively concentrated, ranging from 3.32 to 21.19 MPa. Using the least square method to fit these data in this study, the functional relationship between the elastic modulus and density is obtained, according to Equation (9). It can be concluded that the elastic modulus is an exponential function relationship with the density, and the coefficient of determination ($R^2$) is 0.76, indicating a good correlation between both.

$$E = 0.396 \exp(0.007\rho) \tag{9}$$

where $E$ is the elastic modulus of the snow sample (MPa) and $\rho$ is the density of the snow (kg/m$^3$).

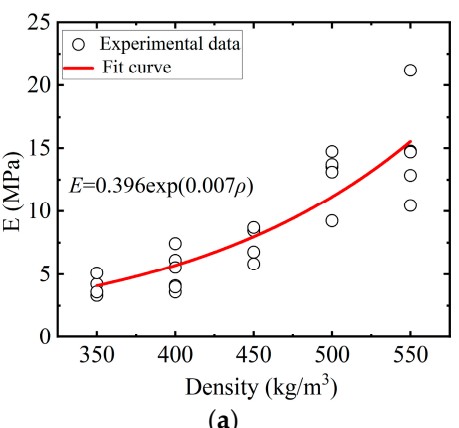 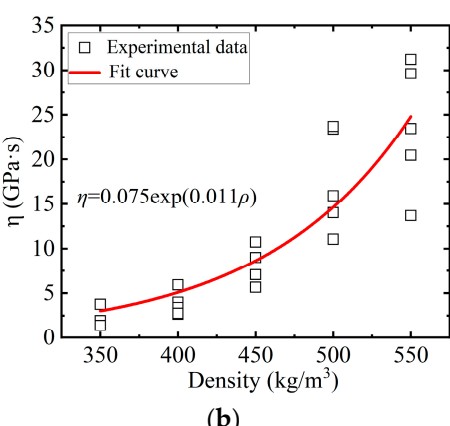

(**a**)　　　　　　　　　　　　　　　　　　　　　(**b**)

**Figure 7.** The relationship between the viscoelastic properties and density: (**a**) the elastic modulus; (**b**) the viscosity coefficient.

As shown in Figure 7b, it is observed that the relationship between the viscosity coefficient and density of snow is similar to that between the elastic modulus and density, both exhibiting an increasing trend with increasing density. In the range of 350 ~ 550 kg/m$^3$, the viscosity coefficient of snow varies from 1.38 to 31.22 GPa·s, which has a relatively poor repeatability compared with the elastic modulus. Furthermore, as the snow density gradually increases, the dispersion of the viscosity coefficient also increases. Taking the results for 350 kg/m$^3$ and 550 kg/m$^3$ as examples, the viscosity coefficient of the 350 kg/m$^3$ snow is from 1.38 to 3.75 GPa·s, and the difference is 2.37 GPa·s. The viscosity coefficient of the 550 kg/m$^3$ snow is from 13.68 to 31.22 GPa·s, and the difference is 17.54 GPa·s. This indi-

cates that the difference in the viscosity coefficient of snow becomes larger with increasing density, emphasizing the need to consider multiple measurements and calculate the average value for accurate results in practical applications. The functional relationship between the viscosity coefficient and density is obtained by the least square method, according to Equation (10). The relationship between the viscosity coefficient and the density of snow is an exponential function, and the coefficient of determination ($R^2$) is 0.78, indicating a good correlation between both.

$$\eta = 0.075 \exp(0.011\rho) \tag{10}$$

where $\eta$ is the viscosity coefficient of the snow sample (GPa·s) and $\rho$ is the density of snow (kg/m$^3$).

This study has simplified snow as a viscoelastic material and neglected the plastic effect of ice grains, employing a simple Maxwell model to develop constitutive models for snow. The experimental results show that despite the presence of multiple variables such as loading rate, temperature, and density, the elastic modulus and viscosity coefficient of snow exhibits a significant exponential functional relationship only with density. It has incorporated the relationship between elastic modulus and density (Equation (9)) and the relationship between viscosity coefficient and density (Equation (10)) into the Maxwell constitutive equation (Equation (5)). These results are applied in a constitutive model (Equation (11)) for natural snow that considers the influence of density. The $R^2$ values of both fitted equations are above 0.70, indicating that the fitted curves are closer to the actual data and the derived constitutive equations are more accurate.

$$\dot{\varepsilon} = \frac{\dot{\sigma}}{0.396 \exp(0.007\rho)} + \frac{\sigma}{0.075 \exp(0.011\rho)} \tag{11}$$

where $\rho$ is the density of snow in which 350, 400, 450, 500, and 550 kg/m$^3$ are taken in the article; $\sigma$ is the compressive strength of snow (MPa); and $\dot{\varepsilon}$ is the strain rate (s$^{-1}$).

## 5. Discussion

### 5.1. Elastic Modulus

Indeed, the effect of various external environmental variables on the elastic modulus of snow samples has received limited attention in previous research [33,34]. The relationship between the elastic modulus and the loading rate obtained in this study is compared with those in the literature [15,35,36]. It is found to be the same as previous reports from [15,36], yet the opposite in conclusion to that obtained in [35]. Lintzén and Edeskär [36] found that the elastic modulus of both artificial snow materials is independent of the loading rate. They performed uniaxial compression tests on coarse-grained snow and fine-grained snow to calculate elastic modulus at the test temperature of $-10\,^{\circ}\text{C}$ and loading rates from 0.5 to 40 mm/min. Scapozza and Bartelt [15] conducted triaxial compression tests at $-12\,^{\circ}\text{C}$ on fine-grained dry snow and found the same trend. On the other hand, Kry [35] performed uniaxial compression tests on alpine snow samples with different densities and strain rates (250~450 kg/m$^3$; $1 \times 10^{-4}$~$2 \times 10^{-3}$ s$^{-1}$). A weak correlation between normalized Young's modulus and strain rate was found, indicating that the elastic modulus of snow increases with strain rate. This result was attributed to the relationship between the elastic modulus of ice and the strain rate. In conclusion, this discrepancy may be attributed to differences in the microstructure of the snow samples, as well as variations in test conditions and methods. To compare and verify these results, standardized test methods for evaluating the viscoelastic properties of snow are required.

Figure 8 compares the elastic modulus versus density for this and other papers [31,37–40]. The results show that the elastic modulus of data F is in the same trend as data B and data D in exponential increase with increasing density. It is worth noting that data B, C, E, and F represent measurements obtained via uniaxial compression test under quasi-static conditions, where the values are relatively concentrated and range from 0.2 to 200 MPa. According to this trend, the value of data F is smaller than that of the other three data,

which may be because step loading includes uniaxial compression and uniaxial creep tests. During the creep test, the stress remains unchanged, and there is a certain time for the rearrangement of the snow grains to achieve a relatively dense spatial structure. Compared with the normal uniaxial compression test, greater deformation is generated under the same stress, so that the obtained elastic modulus is smaller. The data A from [37] represent the dynamic modulus obtained by uniaxial compression tests at high strain rates ($3 \times 10^{-3} \sim 2 \times 10^{-2}$ s$^{-1}$), which is much higher than the elastic modulus value of other data. Köchle and Schneebeli [40] utilize X-ray microcomputer tomography to construct the three-dimensional microstructure of snow and calculate the elastic modulus D using finite element simulation, ranging from 0.3 to 1000 MPa. This value is also higher than the elastic modulus obtained using quasi-static measurements at the same density. This is because the quasi-static measurement of the elastic modulus inevitably takes the viscous strain into account, which leads to the measured elastic modulus not representing the true value.

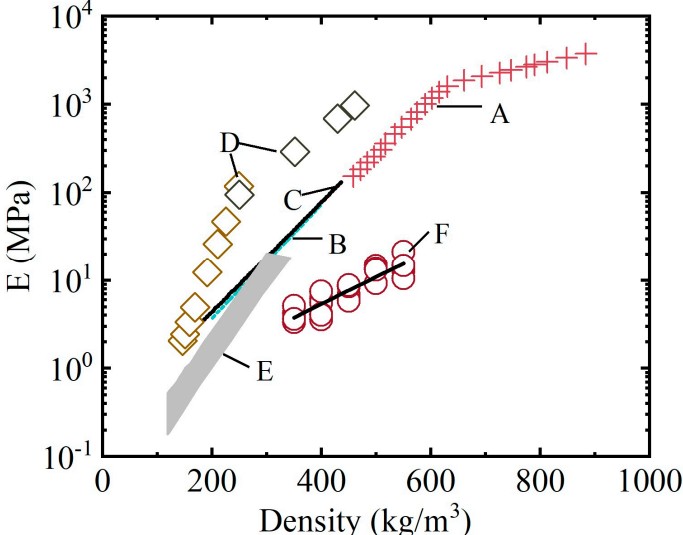

**Figure 8.** The relationship between the elastic modulus and density from different studies: (A) Uniaxial compression, $-25\ ^\circ$C [37]; (B) triaxial compression, $-20\ ^\circ$C $<$ T $< -2\ ^\circ$C [38]; (C) triaxial compression, $-12\ ^\circ$C [41], published in [39]; (D) finite element simulation, the data fit curve [40]; (E) uniaxial compression and tension, $-25\ ^\circ$C $<$ T $< -12\ ^\circ$C [31]; (F) uniaxial compression under step loading, $-5\ ^\circ$C (in this study).

*5.2. Viscosity Coefficient*

Observations of the relationship between the viscosity coefficient and the density of snow showed an exponential growth trend, which is consistent with the results of other studies. For example, Kojima [42] conducted a quasi-static creep test on natural snow with densities ranging from 150 to 350 kg/m$^3$ at $-7\ ^\circ$C to $-10\ ^\circ$C and found a positive correlation between the viscosity coefficient and density. It was concluded that the viscosity coefficient gradually increased with increasing density through triaxial compression tests and uniaxial compression creep tests from [15,35].

The results obtained in this study show an exponential relationship between the elastic modulus and the density of natural snow. Linear regression analysis was used to approximately quantify whether relations existed between elastic modulus and loading rate; however, no tendency for dependence was observed. The functional dependency of the moduli toward temperature was also not observed. Similarly, the viscosity coefficient also increases exponentially with increasing density, and first decreases and then increases with decreasing temperature, but its relationship with loading rate is not clear. Based on these findings, we use the Maxwell model as a framework to present a constitutive equation that take into account the density.

In order to increase and improve the reliability of the constitutive equation, field tests of snow using uniaxial compression equipment may be conducted in future studies. Some field penetration tests have become popular in recent years. Zhao et al. [43] used an Improved Motor-driven Snow Penetrometer to measure the hardness of seasonal snow in Northeast China. The influence of different variables on snow hardness was analyzed using orthogonal tests. Zhuang et al. [44] conducted 74 penetration experiments on seasonal snow in Harbin, China, using a modified Rammsonde. The penetration strength of snow was determined, and its influencing factors were discussed. However, the hardness and the compressive strength obtained from the compression test are two different mechanical parameters and cannot be directly compared.

Snow is made up of countless ice crystals with irregular shapes and sizes. The mechanical properties of snow and ice are similar, but there are also some differences. The elastic modulus of ice increases with decreasing temperature [45]. However, in the present study, the decreasing and then increasing elastic modulus of snow with decreasing temperature was observed only at temperatures greater than 500 kg/m$^3$. Additionally, the tendency of the elastic modulus to change with density is only indirectly affected at higher temperatures ($-5\,°C$ and $-10\,°C$). The effect of temperature on the elastic modulus of snow is relatively complex [46]. Furthermore, the test also shows that the viscosity coefficient of snow increases exponentially with increasing density, first decreasing and then increasing with temperature. Limited by the test conditions, the variable range set in this test was relatively small. For example, the temperature range was only $-5\sim-20\,°C$, the density range was only 350 kg/m$^3$~550 kg/m$^3$, and the loading rate range was only 5~50 N/s. Although the results showed that the elastic modulus and viscosity coefficient varied with temperature and density. A reasonable physical model could not be built to describe the response of elastic modulus and viscosity coefficient to different influencing factors. Therefore, the subsequent temperature and density range should be expanded in an effort to explore its physical phenomenon, which is beneficial for the application of snow in different aspects, for example, in the construction of snow roads in cold regions, runways, and polar infrastructure. The constitutive equations are used to describe the deformation behavior of snow under different loading conditions, which can provide important mechanical parameters and technical support for the design, construction, and maintenance of these snow-related projects.

## 6. Conclusions

In order to investigate the relationship between the elastic modulus and viscosity coefficient of snow with temperature, density, and loading rate, uniaxial compression tests under step loading were adopted in this study. The constitutive equations of snow were further derived by combining the Maxwell model and the functional relationship between viscoelastic properties and different factors. For practical engineering problems represented by polar airport runways, this study can provide some valuable information for their design, construction, and building. Specifically, the major findings in this study are as follows:

1.  The elastic modulus of natural snow increases exponentially with increasing density. Temperature has a certain influence on the elastic modulus of snow. The elastic modulus first decreases and then increases with decreasing temperature, and this relationship is more obvious at high densities. There is no correlation between the elastic modulus and the loading rate.
2.  Density is an important factor in the change in the viscosity coefficient of snow. The viscosity coefficient increases exponentially as density increases. The snow viscosity coefficient is affected by temperature, that is, the viscosity coefficient first decreases and then increases as the temperature decreases. The loading rate is weakly correlated with the viscosity coefficient of snow.

3. A new constitutive equation considering snow density is derived by introducing the functional relation between elastic modulus, viscosity coefficient, and the density of snow based on the Maxwell model.

**Author Contributions:** Conceptualization, H.H. and E.W.; methodology, H.H. and X.L.; formal analysis, H.H. and E.W.; investigation, M.Y., X.L., Y.L. and G.G.; data curation, M.Y., X.L., Y.L. and G.G.; writing—original draft preparation, H.H. and M.Y.; writing—review and editing, X.L. and E.W.; funding acquisition, H.H. All authors have read and agreed to the published version of the manuscript.

**Funding:** This research was supported by the Natural Science Foundation of Heilongjiang Province of China (No. LH2020E004), the Major Scientific and Technological Projects of the Ministry of Water Resources of China (No. SKS-2022017), and the Project to Support the Development of Young Talent by Northeast Agricultural University.

**Data Availability Statement:** The data are available upon request.

**Conflicts of Interest:** The authors declare no conflict of interest.

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
