# Peer review of "Study on the Constitutive Equation and Mechanical Properties of Natural Snow under Step Loading"

_water, doi:10.3390/w15183271_

Round 1

Reviewer 1 Report

1.     Some sentences are long and should be broken, as seen in lines 14-18.

2.     The objectives of the manuscript are not clear. Please define the study's exact goals and your contribution to the literature.

3.     The quality of the figures is poor.

4.     In the first paragraphs of the Introduction, you can include some lines about the significance of snowfall in various regions, such as the Middle East, Andes, Tibetan Plateau, Western US, Canada, and the Alps. You might also want to cite the following sources:

·        Le Roux, E., Evin, G., Eckert, N., Blanchet, J., Morin, S., 2021. Elevation-dependent trends in extreme snowfall in the French Alps from 1959 to 2019. The Cryosphere 15, 4335-4356.

·        Huning, L.S., Margulis, S.A., 2017. Climatology of seasonal snowfall accumulation across the Sierra Nevada (USA): Accumulation rates, distributions, and variability. Water Resour Res 53, 6033-6049.

·        Wang, L., Yang, H., 2023. Tibetan Plateau increases the snowfall in southern China. Sci Rep 13, 12796.

·        Nouri, M., Homaee, M., 2021. Spatiotemporal changes of snow metrics in mountainous data-scarce areas using reanalyses. J Hydrol 603, 126858.

5.     In the conclusion section, please mention your major contribution to the literature.

The Quality of English language can be improved.

Reviewer 2 Report

The paper is an interesting read and might be a useful addition to literature related to compacted snow mechanics. I would recommend the following for the benefit of potential readers.

1. In line 147 on page 4, the word 'set' should be changed to 'setting'.

2. On page 5, maybe a short explanation could be added for the benefit of the readers where the different equations for the Maxwell model are linked together and explained in the text. I appreciate the authors actually including the derivation for the Maxwell model.

3. In line 233 on page 7, maybe it is beneficial for the reader if the point about no correlation between viscosity coefficient and loading rate is elaborated more.

4. The development of the Maxwell model based on density is a nice attempt. It would be beneficial for the readers if the authors also comment on the validity of the equation in case of field testing done with uniaxial compressive devices like the 'Light Weight Deflectometer' or the 'Clegg Impact Hammer'.

Round 2

Reviewer 1 Report

All of my comments have been addressed. Nonetheless, the subpar quality of the figures persists. I believe this issue may stem from the PDF generation process by MDPI, rather than being attributed to the quality of the figures initially provided by the authors.
